# Rab6A as a Pan-Astrocytic Marker in Mouse and Human Brain, and Comparison with Other Glial Markers (GFAP, GS, Aldh1L1, SOX9)

**DOI:** 10.3390/cells10010072

**Published:** 2021-01-05

**Authors:** Linda Melzer, Thomas M. Freiman, Amin Derouiche

**Affiliations:** 1Institute of Anatomy II, Goethe-University, D-60590 Frankfurt am Main, Germany; Linda.Spiess93@gmx.de; 2Department of Neurosurgery, Rostock University Medical Center, D-18055 Rostock, Germany; Thomas.Freiman@med.uni-rostock.de

**Keywords:** trans-Golgi network, astrocyte heterogeneity, immunocytochemistry

## Abstract

Astrocytes contribute to many higher brain functions. A key mechanism in glia-to-neuron signalling is vesicular exocytosis; however, the identity of exocytosis organelles remains a matter of debate. Since vesicles derived from the trans-Golgi network (TGN) are not considered in this context, we studied the astrocyte TGN by immunocytochemistry applying anti-Rab6A. In mouse brain, Rab6A immunostaining is found to be unexpectedly massive, diffuse in all regions, and is detected preferentially and abundantly in the peripheral astrocyte processes, which is hardly evident without glial fibrillary acid protein (GFAP) co-staining. All cells positive for the astrocytic markers glutamine synthetase (GS), GFAP, aldehyde dehydrogenase 1 family member L1 (Aldh1L1), or SRY (sex determining region Y)-box 9 (SOX9) were Rab6A^+^. Rab6A is excluded from microglia, oligodendrocytes, and NG2 cells using cell type-specific markers. In human cortex, Rab6A labelling is very similar and associated with GFAP^+^ astrocytes. The mouse data also confirm the specific astrocytic labelling by Aldh1L1 or SOX9; the astrocyte-specific labelling by GS sometimes debated is replicated again. In mouse and human brain, individual astrocytes display high variability in Rab6A^+^ structures, suggesting dynamic regulation of the glial TGN. In summary, Rab6A expression is an additional, global descriptor of astrocyte identity. Rab6A might constitute an organelle system with a potential role of Rab6A in neuropathological and physiological processes.

## 1. Introduction

Astrocytes, in addition to their metabolic and homeostatic functions, specifically contribute to neuronal network operation and physiological functions in naturally occurring behaviors [1,2,3,4,5,6,7,8]. A key mechanism in signalling from the glial to the neuronal domain is vesicular exocytosis, mostly involving small molecule “gliotransmitters” such as glutamate and d-serine [9]. Moreover, astrocyte-derived proteins, such as S100β, are released to the extracellular space to modulate neuronal firing [10]. Although exocytotic organelles have in fact been observed in astrocytes in situ [11,12,13,14], their identity has remained a matter of debate [15,16,17,18,19]. In particular, there is no general idea as to what extent astrocytic exocytosis organelles occur in brain, or what their biogenesis is. Vesicles derived from the trans-Golgi network (TGN) have not been considered in this context. Moreover, there are very few if any published data on the TGN in astrocytes. We studied the possible presence and distribution of TGN in astrocytes in situ by immunocytochemistry using the TGN marker Rab6A. Although widely studied in cell biology [20], Rab6 is relatively unknown in the field of neuroscience.

Rab6 belongs to the superfamiliy of Ras-GTPases and has its main subtypes A and B [21] with two genes localized on separate chromosomes [22]. Rab6 proteins are involved in the regulation of intracellular membrane traffic [20]. Members of the Rab6 family of proteins localize to the TGN, the Golgi cisterns, and the tubulovesicular organelles moving along the plus end of microtubuli [23]. Rab6 plays a role in retrograde endosome to Golgi transport, it also mediates the Golgi to ER vesicle trafficking [20,24]. Rab6B is regarded as the neuronal Rab6 subtype within the CNS [20,25], but there is no information as to the presence or localization of Rab6A in the CNS.

We thus studied the possible presence of Rab6A in mouse and human brain. We find it in all cells labelled by the astrocytic markers glutamine synthetase (GS), glial fibrillary acid protein (GFAP), aldehyde dehydrogenase 1 family member L1 (Aldh1L1), and SRY (sex determining region Y)-box 9 (SOX9), and it is excluded from other glial cell types.

## 2. Materials and Methods

### 2.1. Tissues

The mouse data reported are based on the brains from 5 adult (10–12 weeks) C57BL/6 mice. Animal handling and sacrifice were according to German government law. Animals were killed by an overdose of isofluorane and perfused through the ascending aorta with a brief rinse (0.9% saline) followed by fixative (4% paraformaldehyde in 0.1M phosphate buffer (PB; pH 7.3)) for 15 min. The brains were taken out and immersion-fixed in the same fixative to complete an overall fixation time of 2 h and rinsed several times in PB. Cryostat sections (12–14 µm), prepared from the brains of 2 mice, were used for NG2 labelling. All other labellings were carried out on 100 µm vibratome sections.

Human tissue from temporal or frontal cortex was obtained from 3 patients during epilepsy surgery (Table 1). The specimens studied here were from surgical access path, or from peripheral portion of resection, and considered largely non-affected by disease. Written consent was obtained from all patients, and the study was approved by the local ethics committee (Medical Faculty, Goethe-Universität Frankfurt am Main, code no. 4/09; project no. SNO-09-2014). The tissue was placed in fixative solution (as above) immediately after resection, cut into 4 mm slabs, and fixed for a total of 16 h. Neuropathological diagnosis was performed by the Department of Neuropathology, Goethe University, University Hospital, Frankfurt am Main, Germany.

### 2.2. Immunocytochemistry

All solutions were prepared in PB. Animal tissues were first incubated for 30 min with 1% NaBH_4_ at room temperature, rinsed for at least 1 h, and then rinsed with mouse Ig-blocking reagent (3%, M.O.M.) for 1 h. After another brief rinse, sections were treated for 30 min with normal horse serum (10%). Incubation with primary antibodies was in 1% normal horse serum, overnight, and at 4 °C (for antibodies and concentrations, see Table 2A). Twenty minutes of rinsing preceded reaction with secondary antibody solution (for secondary antibodies and concentrations, see Table 2B). In case of a biotinylated secondary antibody, sections were then rinsed for another 20 min and incubated for 1 h with fluorochrome-conjugated streptavidin. Nuclear staining was performed with bisbenzimidine. The sections were rinsed for 10 min and coverslipped in fluo-escence mounting medium (Aqua Poly Mount, Polysciences Inc., Warrington, PA, USA).

100 µm thick vibratome sections were prepared from the human cortex tissue slabs and immunoreacted for Rab6A and GFAP, as above, unless otherwise stated. In detail, this included sequential incubations in normal horse serum, anti-Rab6A (overnight, 4 °C), donkey-anti-mouse-Cy3, anti-GFAP Al488 (overnight, 4 °C), and coverslipping.

Sections untreated except for NaBH_4_ were used as controls to assess autofluorescence. To control for specificity of the detection system in Rab6A single staining, sections were immunoreacted without the primary antibody. For a given multiple staining, the series of controls consisted of stainings each omitting only 1 of the 2 or 3 primary antibodies, thus controlling for fluorescence filter bleedthrough, background, and antibody crossreactivities. The results reported were obtained at least in 3 independent staining runs. An overview of the staining combinations carried out in mouse and human tissue is given in Table 3.

### 2.3. Microscopy and Blinded Colocalization Analysis

Control and stained specimens were completely screened, and cells were documented using an upright microscope (Axioskop 2, Zeiss, Oberkochen, Germany) equipped with halogen illumination (HA 100, Zeiss) and narrow-band fluorescence filter sets (472/30/495 DC mirror/513/17 (green channel), 560/25/585/620/60 (red channel), and 406/15/425/460/50 (blue channel)). Images were taken at 10×, 20× (Plan-apo), or 100× (N.A. 1.3, oil) using a 2048 × 2048 pixel monochrome camera (Spot Insight 4, Diagnostic Instruments, Sterling Heights, MI, USA) with 7.45 µm square pixels and 14 bit sampling depth (CCD chip type KAI 4021). Channel overlay was performed at the step of image acquisition using the software Spot 5.0 Advanced (SPOT Imaging Solutions, Diagnostic Instruments), at the same time correcting for chromatic aberration [26].

Uneven image illumination was flattened applying the “rolling ball” algorithm (at 500 pixel radius) as implemented in ImageJ [27]. Background was adjusted on the basis of the control staining, and identically for all frames from a given staining, using Adobe Photoshop. For presentation, brightness and color were adapted to represent the microscope image. Only global, non-selective, and linear image operations were carried out. In some frames of Rab6A/NeuN-stained specimens (see Appendix A), deconvolution was applied to further increase image clarity and resolution, using a motorized microscope (Zeiss Cell Observer) and a CCD (Sony IXC285AL) with 6.45 µm square pixels. Stacks of 60 images were acquired at 50 nm spacing and 100× final magnification (63× 1.4N.A. × 1.6×). The image stacks were deconvolved applying calculated point spread functions (PSFs) and iterative deconvolution (Volocity Software (Quorum Technologies Inc., Puslinch, ON, Canada)). Quantitation of cellular colocalization was blinded to reduce bias for cell morphology and cell type. In double and triple labellings, a reference population was determined corresponding logically to the issue, e.g., to investigate the proportion of Rab6A^+^ microglial cells, the reference population was Iba1^+^, not vice versa. Of the 2 or 3 color channels, all except that of the reference population were rendered invisible in Photoshop. A large number of cells appearing “complete” (with obvious soma, nucleus (presumed, if not stained), and stem processes in the section plane) were first dot-marked on a transparent “overlay” (in Photoshop, see Appendix A). In a second step, the channel to be examined was re-opened for colocalization. Only the dot-marked cells were checked for cellular anti-gen colocalization to finally obtain the percentage of Rab6A^+^/Iba1^+^ microglial cells. In some cases, double and triple labellings were repeatedly examined with different reference populations, to study different questions.

Assessing colocalization at the level of cross-sectioned soma and nucleus largely precludes false positive colocalization by superimposition in 3D. Firstly, absence of tissue permeabilization, as in our protocol, results in antibody penetration depth of less than 1 µm. Also, cells below the section surface can only be labelled intracellularly if continuous with a process physically cut by tissue sectioning; the plethora of small astrocytic processes exposed at the section surface are too narrow to provide antibody access to deeper structures. Second, with a depth of focus of approximately 500 nm (100× objective), and an established astrocyte nucleus size of 5–6 µm, spurious superimpostion of the Rab6A^+^ section surface and a deeper cell soma is highly improbable. In this context, we did not observe Rab6A^+^ cell nuclei. In summary, the images examined for colocalization represent labelling at the cell surface, in a focal plane approximately 500 nm thick.

Distribution of Rab6A^+^ in astrocytes and other cells was assessed qualitatively by structural analysis of “complete cells” in the cortex, corpus callosum, hippocampus, and thalamus, imaged at 100×, altogether 14.568 astrocytes in 1774 frames, from 3 mice. For quantitative assessment, morphometric object-oriented image analysis was performed on regions-of-interest (ROIs) of 10 representative cells each from the 4 types found and classified qualitatively. Using particle analysis and planimetry commands in ImageJ [27], we determined individual area and number of Rab6A^+^ objects.

## 3. Results

### 3.1. Rab6A is Massively Present All Over the Brain

At low magnification, Rab6A staining in mouse brain sections is abundant and diffuse in all areas of forebrain and brainstem, resembling background staining. Autofluorescence in sections reacted without primary antibody is considerably weaker (Figure 1A,C,E) so that Rab6A staining, e.g., in cortex, hippocampus, septum (Figure 1B,D,F), and other regions (e.g., hypothalamus, cerebellum, amygdala, thalamus; Appendix A) is regarded as signal, including white matter (Figure 1B). Signal intensity varies between gray and white matter, and across layers (Figure 1B,D,F). Processes and fluffy patches (arrows) suggest astrocyte territories; moreover, blood vessels (asterisks) are clearly outlined (Figure 1B,D).

### 3.2. Localisation of Rab6A in Astrocytes

At higher magnification, Rab6A staining is grainy, however, not homogeneous, but with “rings” and “alleys”, presumably negatively outlined neuronal dendrites in cross- and longitudinal section (Appendix A). Rab6A^+^ labelling, e.g., in hippocampus, indicates preferential labelling of astroglial structures, with stem processes of fibrous astrocytes in white matter (Appendix A), labelling of ependymocytes (Appendix A), vascular outlines in the hippocampal fissure (Appendix A), and encircling of the granule cells (Appendix A). Continuous and intense cellular label was seen in the subgranular zone of the fascia dentata, where the cells labelled may be subgranular astrocytes and/or stem cells (Appendix A).

Astrocyte labelling could be substantiated in sections double stained for Rab6A and GFAP (Appendix A). Examining GFAP^+^ astrocytes at high resolution, we found that Rab6A^+^ structures interestingly appear as puncta or clusters of puncta. They are partly juxtaposed with GFAP^+^ main processes and somata (Figure 2). Rab6A^+^ structures also appear to fill the numerous GFAP^−^ peripheral astrocyte processes (PAPs) known to emerge from the main processes and to make up the diffuse label at low power (Figure 1B,D and Appendix A). Rab6A^+^ puncta may also be localized in the perivascular glial endfeet (Appendix A).

### 3.3. Rab6A is Contained in All Astrocytes

Blinded colocalization quantitation in GFAP^+^ astrocytes in hippocampus, thalamus, cortex, and corpus callosum showed that 100% of GFAP^+^ astrocytes also display Rab6A staining (Appendix A), demonstrating that GFAP^+^/Rab6A^+^ astrocytes do not represent a subpopulation of GFAP^+^ astrocytes.

It is long established that only a varying proportion of astrocytes display GFAP, in a region-dependent manner [28]. To examine whether all astrocytes or only the GFAP^+^ or other subpopulation displays Rab6A labelling, we combined Rab6A or Rab6A/GFAP staining with immunolabelling for one of three accepted pan-astrocytic markers, i.e., markers present in all astrocytes (Table 3), v.z. GS [29], Aldh1L1 [30,31], or SOX9 [32]. In blinded analysis, all GFAP^+^/Rab6A^+^ astrocytes were co-labelled for the pan-astrocytic markers GS or Aldh1L1 (Figure 3, Appendix A). Using only double labellings and reversing the reference population, we found that 100% of the cells labelled with the pan-astrocytic markers GS, Aldh1L1, or Sox9 are positive for Rab6A (Appendix A; Figure 4 and Appendix A; Appendix A).

Since there is some debate as to the validity of GS or Aldh1L1 as pan-astrocytic markers, we quantitatively assessed the same material with different reference populations. Thus, we studied the proportion of GFAP^+^ astrocytes either in GS^+^/Rab6A^+^ astrocytes or Aldh1L1^+^/Rab6A^+^ astrocytes. GFAP co-staining varies in a region-dependent way that is similar in both triple stainings, i.e., with GS^+^/Rab6A^+^ and Aldh1L1^+^/Rab6A^+^. While all GS^+^/Rab6A^+^ or Aldh1L1^+^/Rab6A^+^ astrocytes in the corpus callosum (fibrous astrocytes) and in the hippocampus display GFAP label, only 58% and 26% of the GS^+^/Rab6A^+^ astrocytes were GFAP^+^ in cortex and thalamus, respectively (Appendix A), and 24% and 14% of the Aldh1L1^+^/Rab6A^+^ astrocytes (Appendix A).

In summary, the double stainings showed that all astrocytes, as defined by labelling for GS, Aldh1L1, or SOX9, display Rab6A in the regions studied. Moreover, the GFAP^+^ subpopulation of astrocytes represents a region-dependent fraction of the Rab6A^+^ astrocyte population, which parallels the established relation of the GFAP^+^ and GS^+^ astrocytes [28,33].

### 3.4. Rab6A is an Astrocyte-Specific Marker

Studying the morphology of Rab6A staining, it appears that Rab6A labels astrocytes only. To verify this, we evaluated microglial cells, oligodendrocytes, NG2-glial cells, and neurons for potential Rab6A labelling in sections double labelled for the cell type-specific markers Iba1, CNPase, NG2, and NeuN, respectively. The non-astrocytic cells constituted the reference population in blinded analysis. All cell types were sampled from cortex, and oligodendrocytes also from corpus callosum and hippocampus; cell numbers are given in Appendix A. Neurons were not counted. Depending on secondary antibody, there was false positive Rab6A staining selectively in microglial cells (1–3 larger granula in many cells) present also in the controls (see Appendix A). Only with biotinylated horse anti-goat (followed by CY3-conjugated streptavidin) and donkey anti-mouse Alexa 647 was this non-specific, selective staining prevented. Virtually all non-astrocytic cells examined are negative for Rab6A: 99% of NG2 cells; 100% of microglia; and 98.3% of oligodendrocytes from cortex, 99.5% from corpus callosum, and 100% from hippocampus (Figure 5), and no Rab6A^+^ neurons were found (Appendix A).

### 3.5. Morphology and Subcellular Distribution of Rab6^+^ Puncta

At high magnification, Rab6A^+^ grains are pleomorphic, and their shape is mostly round and elongated; frequently they are contiguous, thus adding to their overall complex structure (Figure 2C). The smallest Rab6A^+^ structures are at the light microscopic detection limit and are highly abundant, resulting in a “salt and pepper” appearance (Figure 6A–C). As evident in GFAP double staining, the majority of Rab6A^+^ grains predominantly fill the PAPs but are also aligned along the GFAP^+^ stem processes (Figure 6A–C). However, the subcellular distribution, grain size, and number vary from one cell to the other. For descriptive reasons, we divided astrocytes into four types (I–IV) on the basis of their subcellular distribution, morphology, number, and size of Rab6A^+^ structures within the individual cell. The types are described and illustrated in Table 4 and Figure 6, respectively. This classification was confirmed by significant differences in object-oriented morphometry (Appendix A). The Rab6A^+^ grain size in area units (µm^2^; Table 4) is meant to illustrate only relative size, since the actual size of the smaller grains is at the limit of light microscopic resolution and is considered subject to nonlinear distortion.

As a general observation, the number of Rab6A^+^ puncta successively decreases from types I–IV, and they become more “centralized” in the perinuclear cytoplasm, while concomitantly, their size increases, with a pronounced increase in type IV (Figure 6A–D and Appendix A). These astrocyte types I–IV are not region-specific, and their relative frequencies in the thalamus and telencephalon decrease from I–IV (Figure 6E). The distribution of types I–IV within tissue appears scattered and random, and there are no accumulations, and there is no predominant localization with respect to layers, the pial surface, vessels, neurons, or white matter.

### 3.6. Rab6A is Also Contained in Human Astrocytes

We tested the validity of the observations in mouse brain for the human. Freshly fixed human cortex was obtained from three patients during epilepsy surgery (see Table 1) and labelled for Rab6A or Rab6A/GFAP. Like in mouse brain sections, the Rab6A signal is above background autofluorescence, which shows as an even stain all over the section at low magnification, and is granular at high magnification. In Rab6A/GFAP double staining, Rab6A is clearly localized in astrocytes, predominantly in their stem and peripheral processes, where it is frequently clustered, appearing as bunches of grapes (Figure 7).

We applied the classical morphological criteria, v.z. hypertrophy of soma and main processes, as well as GFAP overexpression, as a working definition of reactive astrocytes in human, although it is widely accepted now that astrocyte reactivity is a complex and heterogeneous phenomenon comprising changes in morphology, gene expression, signaling, and proliferation [34]. Thus, both non-reactive and reactive astrocytes were observed to varying degrees in tissues from patient cases I and II, which were diagnosed focal cortical dysplasia (see Table 1). Exclusively non-reactive astrocytes were seen in the cortex tissue removed for surgical access in patient case III (ganglioglioma in hippocampus). Double staining was blindly quantitated with GFAP as the reference channel, examining separately non-reactive and reactive astrocytes. All non-reactive astrocytes examined in cases I-III were Rab6A^+^; of the reactive astrocytes studied, 100% of case II and 88% of case I were Rab6A^+^, leaving 12% of Rab6A^-^ astrocytes (Appendix A).

We further studied whether the classification of mouse astrocyte types based on Rab6A^+^ grain number, size, and distribution within the individual cell is reflected in human astrocytes. In the reactive and non-reactive astrocytes studied, the astrocyte types I–IV were similarly observed and classified, and were found at varying relative frequencies (Appendix A). In summary (Appendix A), relative frequencies in cases I–II did not stepwise decrease over types I–IV, as found in mouse astrocytes. This was the case only in the cortex tissue from case III, which can be assumed not to display pathological alterations, and did not contain reactive astrocytes. In the cortex from patient cases I and II, there was no consistent correlation between the frequencies of types I–IV and reactive vs. non-reactive astrocytes (Appendix A). As in mouse brain, the distribution of astrocyte types I–IV in human cortex tissue appears random; there is no predominant localization with respect to structural features, cortical layers, the pial surface, vessels, or neurons.

## 4. Discussion

### 4.1. Astrocyte Heterogeneity and Cell Type Specificity

All analyses of cellular co-localization carried out here were not by general impression but by blinded and quantitative assessment of a representative number of individual cells. As a major finding of this study, Rab6A is selectively and specifically localized in astrocytes.

First, Rab6A could not be detected in neurons or any of the other glial cell types investigated (microglia, oligodendrocytes, NG2 cells) by double-immunostaining using well-established markers (NeuN, Iba1, CNPase, and NG2).

Second, a series of mutually confirming colocalization studies led to the conclusion that all astrocytes are positive for Rab6A, particularly in the light of extant data on GS and Aldh1L1 labelling. In spite of some studies reporting presence of GS in non-astrocytic cell types, GS may be considered the most inclusive and specific astrocyte marker [28,29]. Aldh1L1 was first described as an astrocytic marker (immunocytochemistry [31]), and this was extended to specific, pan-astrocytic labelling by transcriptome analysis of Aldh1L1-expressing cells [30] using BAC ALDH1L1 eGFP mice. A minor Aldh1L1^+^ subpopulation of oligodendrocytes observed by Yang et al. [35] in the same BAC ALDH1L1 eGFP mice had been excluded by Neymeyer et al. [31] and Cahoy et al. [30]. Analyzing double stainings for Rab6A/GS and Rab6A/Aldh1L1, and using alternate reference populations, all Rab6A^+^ cells were GS^+^ and Aldh1L1^+^, and there were no GS^+^ or Aldh1L1^+^ single-stained cells. This confirms the congruence of GS^+^ and Aldh1L1^+^ populations, at the same time establishing that Rab6A is a pan-astrocytic marker.

There is general consensus that both GS and Aldh1L1 stain more astrocytes than GFAP, in a region specific manner [28,29,30,31,36]. This is in line with the expression of GS or Aldh1L1 in all regions examined here, in 100% of the GFAP^+^/Rab6A^+^ cells. Similarly, only a fraction of GS^+^/Rab6A^+^ or Aldh1L1^+^/Rab6A^+^ astrocytes examined were GFAP^+^ in the cortex and thalamus, thus further confirming extant data and the most inclusive labelling of astrocytes by Rab6A. In addition, the Rab6A/SOX9 double staining would further confirm both the recent introduction of SOX9 as a specific pan-astrocytic marker [32] and the conclusion of Rab6A as such a marker.

Consulting published CNS transcriptome analyses [30,37,38,39] for cell type-specific *RAB6A* expression yields ambiguous results. Rab6 may be present in its main isoforms Rab6A and Rab6B, but these transcriptome studies are based on RNA chips that do not specifically address *RAB6A* transcription; they apply one sequence for *RAB6B*, and another one designated “*RAB6*”, which we assume is non-discriminatory. However, the transcriptome data permit to concluded that the *RAB6B* gene is hardly expressed in astrocytes if at all [30,39], in line with Doyle et al. [37], who could not detect *RAB6B* expression in astrocytes. Further, *RAB6B* expression has been found to be mainly neuronal, since only neurons were consistently positive with the several Rab6 probes used [30], a finding confirmed by the analysis by Zhang et al. [39], and the view that in the CNS, *RAB6B* is the neuronal isoform [20,25]. Thus, while these transcriptome analyses do not specifically confirm the present data, they also do not contradict them.

The Rab6A monoclonal antibody (clone 3G3) used here is raised against a C-terminal peptide with a sequence present only in the Rab6A but not the Rab6B isoform. Its intracellular staining pattern with the dense peri- or juxtanuclear cisterns, vesicles, and radiating puncta described here is typical of distal Golgi cisterns, the TGN, and TGN-derived organelles. In the field of cell biology, this Rab6 pattern is established without debate in numerous cell lines and studies, using other anti-Rab6 antibodies or fluorescent fusion proteins (e.g., [23,24,40], reviewed in [20]). The Rab6A antibody applied here has previously replicated this typical vesicular perinuclear staining in Chinese hamster ovary (CHO) cells [41]. To our knowledge, however, immunocytochemical localization of Rab6A or B in mammalian tissues has been determined only by very few studies (e.g., [25,39,41,42,43], see above), including the present one.

Our study is the first to systematically investigate cell type-specific localization of Rab6A in the CNS. Previous CNS data on Rab6 in general are sparse and not very conclusive. Thus, Huang et al. [44] demonstrated immunolocalization and light-dependent expression of Rab6 in mouse and rat retina. Applying colabelling for GS, the authors concluded that Rab6 is not localized in the GS^+^ Müller cells, without supplying an alternative, although we find that their data would not exclude Müller cells. Studies of Rab6 expression by immunostaining and immunoblot in hippocampus and cortex from patients diagnosed with Alzheimer’s disease revealed pronounced signal in neurons [45]. The polyclonal antibody (anti-Rab6, Santa Cruz) used in that study recognizes all Rab6 isoforms, however, it detects Rab6B at a much lower concentration (1:2000) than Rab6A (1:50) (Derouiche, unpublished). Since Rab6B is regarded as the neuronal isoform [20,25], it can be assumed that Scheper et al. [45] used the antibody at low concentration. Thus, their results would be in line with transcriptome data (see above). In the cerebellum, Rab6B was reported to be specifically expressed in microglia, pericytes, and Purkinje cells, however, by comparing the pattern of Rab6B staining with that of cell type-specific labelling in different sections [25]. The issue of astrocyte identity has been complicated by that of astrocyte heterogeneity, which is multi-dimensional with approaches addressing morphology, function, gene expression, physiology, development, disease, or aging (reviewed by [28,35,46,47,48,49,50,51,52]). Our data do not add to the multiple facets of astrocytes; on the contrary, they supply an additional, global descriptor of astrocyte identity, v.z. expression of Rab6A. Strictly speaking, the data are limited to the mouse and human brain, and do not permit extrapolation to other species, development, aging, or disease. Given the pan-astrocytic and specific distribution of Rab6A, as well as the high phylogenetic conservation of Rab6 [20], it might be speculated that the as yet unknown function of astrocytic Rab6A is crucial for the CNS, and that Rab6A, like GS, is expressed by astrocytes in vertebrate species in general [29,53]. However, this remains the object of future studies.

In the human, all GFAP^+^ non-reactive astrocytes examined were Rab6A^+^. We regard the GFAP^+^/Rab6A^-^ astrocytes found as associated with pathology (see below), since they constituted 12% of reactive astrocytes and were only observed in tissue from patient case I (FCD). We, therefore, suggest that Rab6A is a specific and selective glial marker also in the human, even though double staining for non-astrocytic glial cells was not performed.

### 4.2. Astrocyte Cell Biology

Our astrocyte classification into types I–IV, on the basis of Rab6A, did not result from unbiassed analysis (e.g., cluster analysis), but was post hoc following microscopic inspection. Intracellular Rab6A grains were found to be heterogeneous with respect to subcellular distribution, morphology, size, and number per cell. These parameters appear to be linked, which led to the classification (types I–IV, see Table 4). There were definitely many “transitional” cells that cannot be assigned to one of the types I–IV. We do not propose that astrocyte types I–IV represent discrete glial subtypes; we feel it more appropriate regarding types I–IV as intermediate states of dynamic processes within the cell. For further comment on the astrocyte types I–IV, it is interesting to consider the nature of the Rab6A^+^ structures. Rab6 is known as a marker of the TGN, and in other cell types, Rab6 localizes to late Golgi cisterns, the TGN, as well as TGN-derived cisterns and vesicles [20]. We believe that also in astrocytes, the fluorescent Rab6A grains of varying size and shape are membrane-bound organelles. The larger and medium sized Rab6A^+^ structures observed in perinuclear position and in stem processes may be larger cisterns or cisterns at intermediate stages of fragmentation, respectively, while the very small Rab6^+^ puncta at the limit of microscopic resolution may be vesicles. As part of TGN development, large TGN cisterns originate near the Golgi, break up into smaller cisterns, and give rise to many small vesicles that bud off from the cisterns, thus consuming them [24]. Since Rab6A^+^ structures also “fill” the astrocytic stem and peripheral processes, it is conceivable that astrocytic TGN cisterns may be transported to sites very distant from the soma, where they fragment and bud off small vesicles. In this light, the astrocyte types I–IV, in reverse order, might correspond to TGN development. As a summary observation, the number of Rab6A^+^ puncta successively increased from type VI-I, while their size decreased concomitantly, from the few large, perinuclear Rab6A^+^ “lumps” (type IV) to medium-sized and more frequent small grains (types III-I; Figure 6 and Appendix A). It remains completely unclear as to whether type IV astrocytes and its TGN develop to types III-I, or remain at that stage. In any case, the morphological data show that there is some degree of regulation of astrocytic TGN development, be it intrinsically or by intercellular signaling, for example, by neuronal activity.

It is interesting to note here that the few Rab6A^-^ astrocytes found were reactive and restricted to one of the two cases of FCD. It may be hypothesized that absence or suppression of TGN development in these astrocytes may result from extrinsic, pathology-associated biological mechanisms. Alternatively, Rab6A^-^ astrocytes may represent a particular type of astrocytic reactivity involving loss of general astrocyte features, such as GS expression, which is also downregulated, e.g., in epilepsy [54,55].

The study in human tissue, although limited, is conclusive about the general presence of Rab6A in astrocytes in the human, but the neuropathological findings from three cases are a non-representative, initial pilot study, suggesting that reactive astrocytes are mostly Rab6A^+^, but can be Rab6A^-^. It prompts systematic pathological studies aimed at understanding both the possible role of astroglial Rab6A in pathology and its function in the brain.

Astrocytic Rab6A^+^ TGN organelles are demonstrated here to be ubiquitous and abundant in the brain, and Rab6 is evolutionary highly conserved [20]. Cell biological studies will have to clarify whether these TGN vesicles recycle, for example, to the ER, or are targeted to the plasma membrane for endocytosis and/or exocytosis.

Ongoing studies aim at testing whether Rab6A^+^ TGN-derived vesicles are exocytosed in astrocytes. If so, this might shed a light on the identity of glial organelles related to release of transmitters and proteins involved in glia-neuronal communication [10,15,16,17,18,19], mechanisms known to regulate vital functions [56] and behavior [1,2,3,4,5,6,7,8].

We show here that Rab6A is a pan-astrocytic marker, a marker specific and common to all astrocytes. The data also confirm the global and specific astrocytic labelling by Aldh1L1 in gray matter, and the astrocyte-specific labelling by GS, which is sometimes debated [28,29], was replicated again. In both mouse and human brain, individual astrocytes display a variability in subcellular distribution, size, and number of Rab6A^+^ structures, suggesting dynamic regulation of the glial TGN.

## Figures and Tables

**Figure 1 cells-10-00072-f001:**
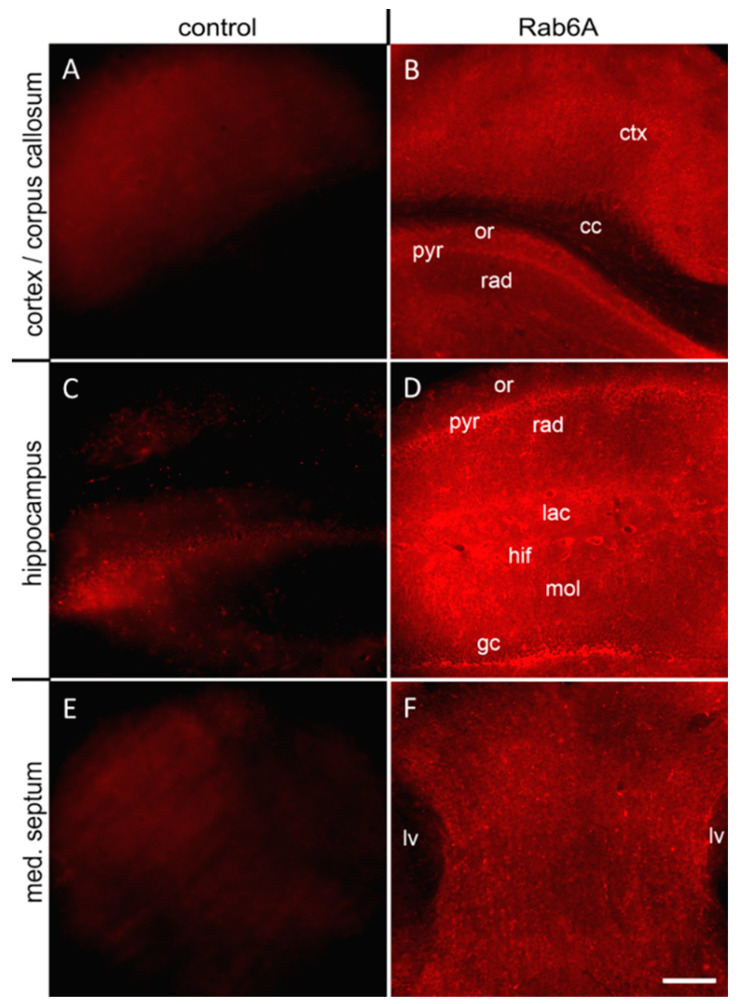
Rab6A is massively present all over the brain. In relation to the corresponding controls reacted without primary antibody (**A**,**C**,**E**), specific immunostaining of mouse tissue sections showed massive, uniform, and ubiquitous presence of Rab6A in all brain regions investigated; including (**B**) cortex with corpus callosum, (**D**) hippocampus, (**F**) medial septum. Note blood vessels in hippocampal fissure (hif). cc, corpus callosum; ctx, cortex; gc, granule cell layer; hif, hippocampal fissure; lac, stratum lacunosum-moleculare; med. septum, medial septum; mol, molecular cell layer; or, stratum oriens; pyr, stratum pyramidale; rad, stratum radiatum. Scale (in **F**, for **A**–**F**): 50 µm.

**Figure 2 cells-10-00072-f002:**
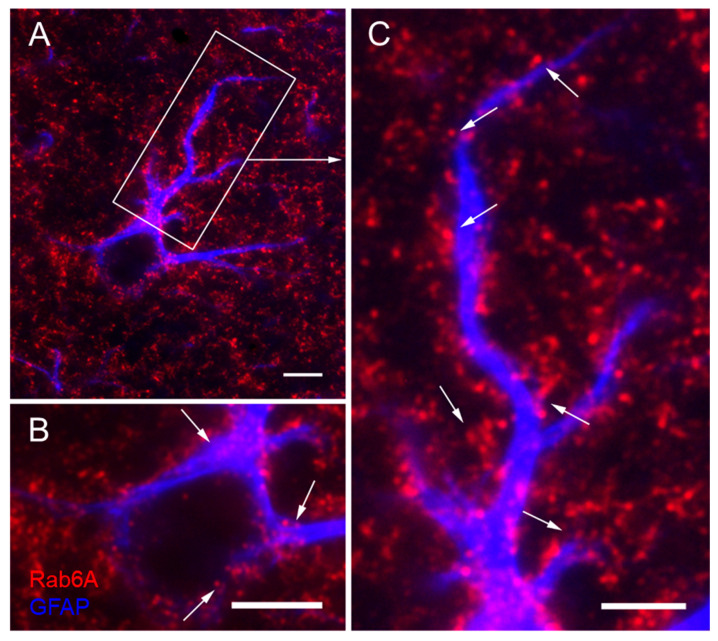
Morphology and subcelluar distribution of Rab6A in glial fibrillary acid protein (GFAP)^+^ astrocytes. Countless Rab6A^+^ grains (**A**) resolve to single puncta or chains of puncta at higher magnification ((**B**,**C**), from (**A**)) that may be associated with GFAP^+^ processes (**A**) or soma (arrows in (**B**)). Rab6A^+^ chains of puncta appear to fill the GFAP^−^ fine processes that are connected to the main processes (arrows in (**C**)). Hippocampus, stratum radiatum of CA1. Scale: 5 µm (**A**,**C**), 3 µm (**B**).

**Figure 3 cells-10-00072-f003:**
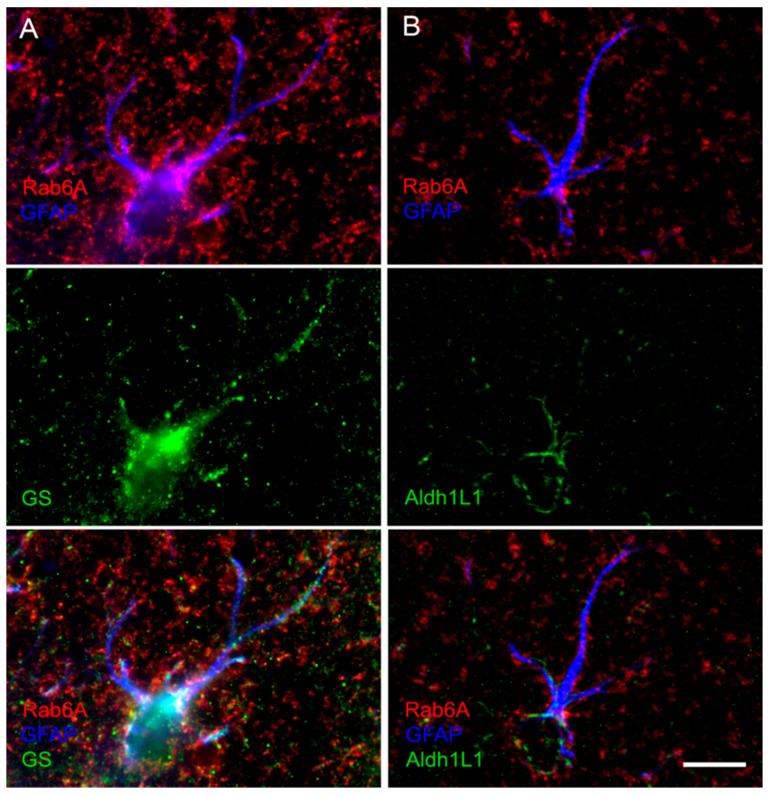
Glutamine synthetase (GS) and aldehyde dehydrogenase 1 family member L1 (Aldh1L1) are localized in all GFAP^+^/Rab6A^+^ astrocytes. All GFAP^+^/Rab6A^+^ astrocytes assessed for either GS or Aldh1L1 were triple-stained (see Appendix A). Note the pronounced punctate GS/Rab6A double staining in the abundant GFAP^-^ peripheral processes ((**A**), merge). This is less pronounced in triple staining with Aldh1L1 (**B**), where labelling is distinct in soma. Hippocampus, stratum oriens (**A**), and radiatum (**B**). Scale: 10 µm (for (**A**,**B**)).

**Figure 4 cells-10-00072-f004:**
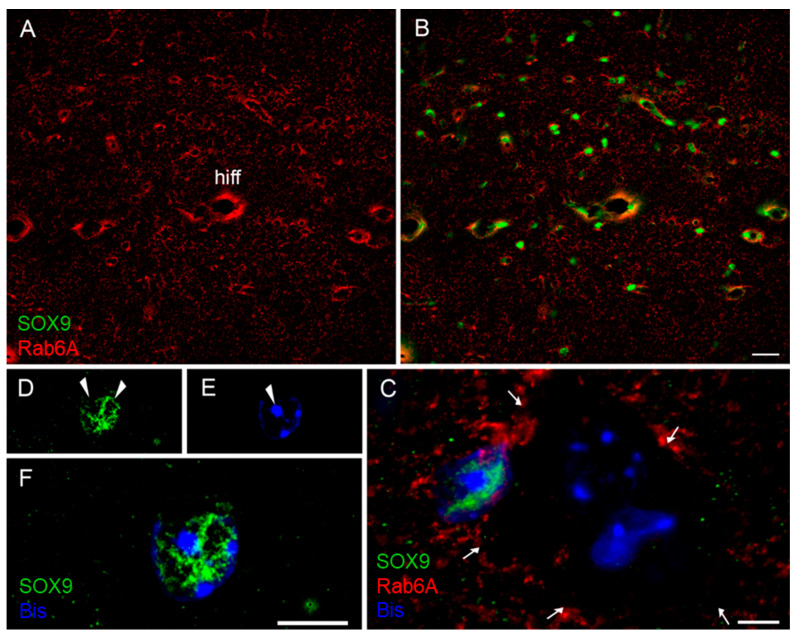
All Sox9^+^ cells are Rab6A^+^. All openings for nucleus visible in Rab6A staining (**A**) are filled by SOX9 staining (**B**) (for quantitation, see Appendix A). (**C**) The nucleus of a neuron demarcated by Rab6A^+^ glial processes (arrows) is SOX9^−^, whereas that of a perineuronal astrocyte is SOX9^+^. Within astrocyte nuclei, gaps in SOX9 label ((**D**), arrowheads) and heterochromatin ((**E**), bisbenzimidine, arrowheads) are complementary (**F**). hiff, hippocampal fissure; Bis, bisbenzimidine. Scales: 50 µm (for (**A,B**)), 5 µm (**C,F**).

**Figure 5 cells-10-00072-f005:**
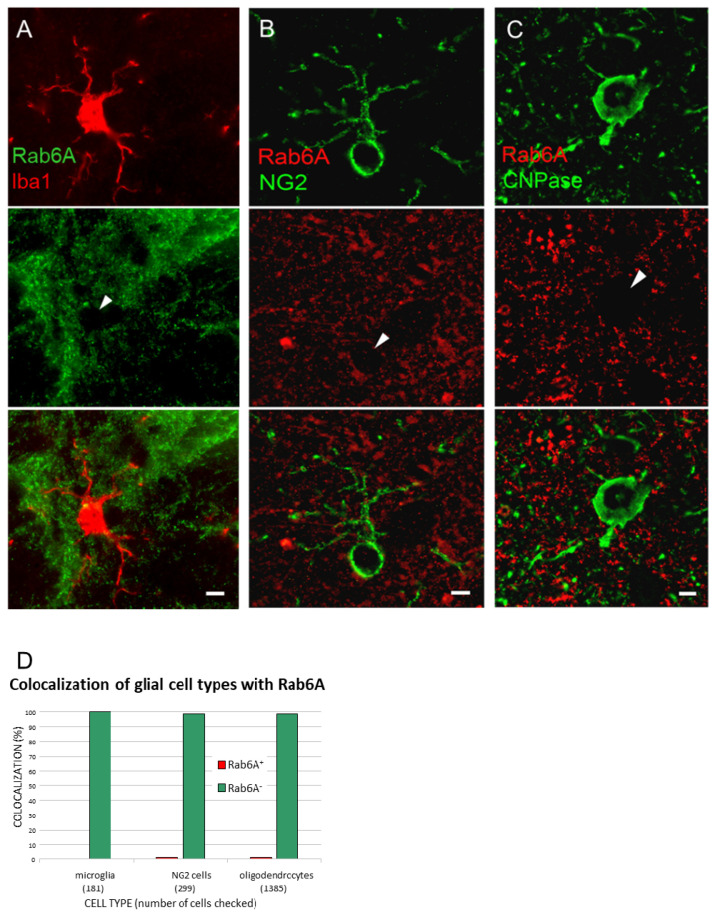
Rab6A is specifically associated with astrocytes. In blinded colocalization analysis in preselected cells stained for Iba1 ((**A**) microglia), NG2 ((**B**) NG2 cells), or CNPase ((**C**) oligodendrocytes), nearly all cells are gaps in the Rab6A channel (arrowheads in (**A**–**C**)), and Rab6A- in double staining (bottom row). (**D**) Quantitation of Rab6A localization glial cell types. Scales: 10 µm (**A**), 5 µm (**B**,**C**).

**Figure 6 cells-10-00072-f006:**
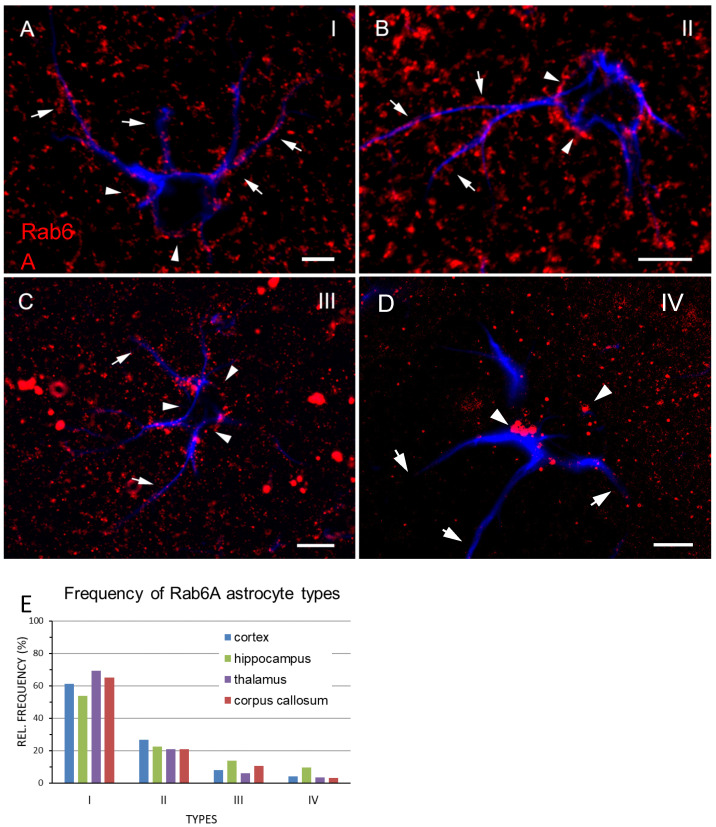
Classification of Rab6A^+^ astrocyte types I–IV. (**A**) Type I—evenly sized Rab6A^+^ puncta are distributed all over the perinuclear region (arrowheads) and in stem and peripheral processes (arrows). (**B**) Type II—Rab6A^+^ puncta in the perinuclear region (arrowheads) and in stem and peripheral processes (arrows) vary in size and may occur in clusters. (**C**) Type III—There is irregular distribution and high variability in size of Rab6A^+^ puncta in the perinuclear region (arrowheads). In stem and peripheral processes (arrows), Rab6A^+^ puncta are evenly but sparsely distributed. (**D**) Type IV—the perinuclear region contains Rab6A^+^ puncta of highly varying size occurring in clusters, mostly on one side of the nucleus (arrowheads). Hardly any Rab6A^+^ puncta are present in the processes (arrows). (**E**) Relative frequencies of types I–IV; percentage values refer to 100% of each type. Scales: 5 µm (**A**,**D**), 10 µm (**B**,**C**).

**Figure 7 cells-10-00072-f007:**
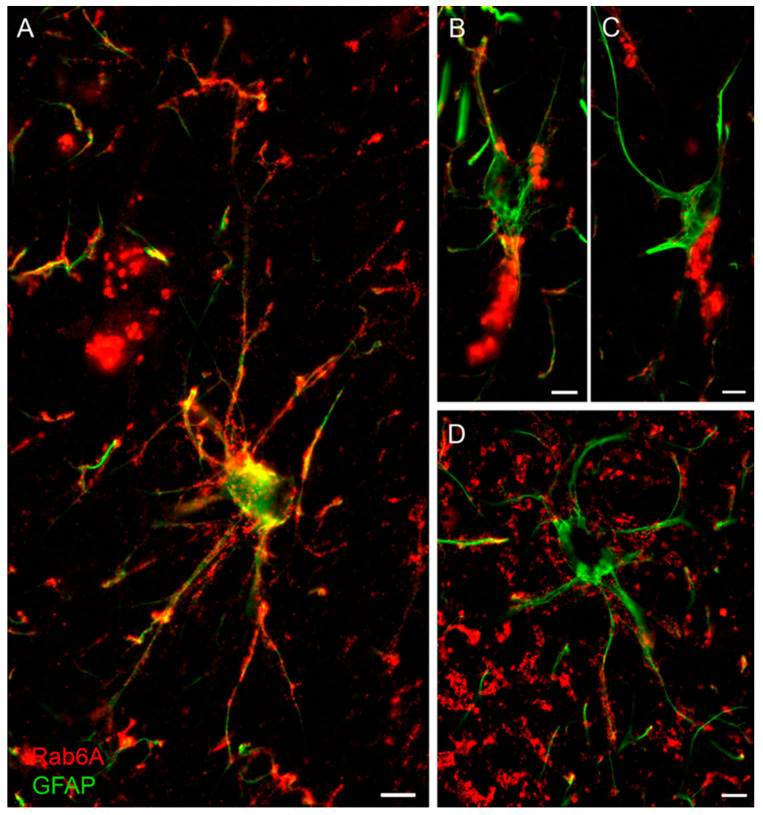
Rab6A in human GFAP^+^ astrocytes. (**A**–**D**) Examples from three cases. All of the preselected human non-reactive GFAP^+^ astrocytes and most of the reactive GFAP^+^ astrocytes were Rab6A^+^ (for quantification, see Appendix A). Note variability of cellular Rab6A staining with pronounced label in the distal processes (**A**,**D**) or predominant staining in the perinuclear region (**B**,**C**). Scales: 10 µm (**A**,**D**), 3 µm (**B**), 5 µm (**C**).

**Table 1 cells-10-00072-t001:** Patient cases.

Patient Case	Age (y)	Sex	Neuropathological Diagnosis	Tissue Studied
I	31	m	focal cortical dysplasia (Palmini II, ILAE IIa) temporal cortex and amygdala, hippocampal sclerosis (ILAE III)	temporal cortex
II	15	m	focal cortical dysplasia (ILAE IIb), frontal cortex	frontal cortex
III	20	m	ganglioglioma WHO grade I with hippocampal infiltration (grade of sclerosis not rated)	cortex from access path

**Table 2 cells-10-00072-t002:** Primary antibodies.

A			
Antibody	Host	Supplier, Cat#, RRID ^1^	Concentration
Aldh1L1	rabbit	Sigma-Aldrich, HPA036900, RRID:AB_10672273	1:200
CNPase	rabbit	Synaptic Systems, 355 002, RRID:AB_2620111	1:500
GFAP	chicken	Millipore, AB5541, RRID:AB_177521	1:500
GFAP Al488	mouse	Cell Signaling Technology, 3655, RRID:AB_2263284	1:100
GS	goat	Santa Cruz Biotechnology, sc-6640, RRID:AB_641095	1:500
Iba1	goat	Abcam, ab5076, RRID:AB_2224402	1:1000
NeuN	guinea pig	Synaptic Systems, 266004, RRID:AB_2619988	1:2000
NG2	rabbit	Millipore, AB5320, RRID:AB_91789	1:500
Rab6A ^2^	mouse	Sigma-Aldrich, WH0005870M1, RRID:AB_1843236	1:5000
Sox9	goat	R and D Systems, AF3075, RRID:AB_2194160	1:500
**B**			
**Antibody/Reagent**		**Supplier**	**Concentration**
Donkey-anti-chicken-AMCA		Jackson Immunoresearch	1:100
Donkey-anti-rabbit-Alexa 488		Jackson Immunoresearch	1:100
Donkey-anti-mouse-Alexa 488		Jackson Immunoresearch	1:100
Donkey-anti-mouse-Alexa 647		Jackson Immunoresearch	1:100
Donkey-anti-mouse-Cy3		Jackson Immunoresearch	1:1000
Donkey-anti-goat-Dy 488		Jackson Immunoresearch	1:100
Donkey-anti-guinea pig Alexa 488		Jackson Immunoresearch	1:100
Horse-anti-rabbit-biotin		Vector Laboratories	1:217
Horse-anti-mouse-biotin		Vector Laboratories	1:217
Horse-anti-goat-biotin		Vector Laboratories	1:217
Bisbenzimidine		Sigma	1:2000
Mouse-Ig-Blocking Reagent (M.O.M.)		Vector Laboratories	3%
Normal horse serum		Vector Laboratories	1%, 10%
Streptavidin-Al488		Vector Laboratories	1:100
Streptavidin-Cy3		Vector Laboratories	1:1000

^1^ Antibody reference according to the Resource Identification Initiative at https://scicrunch.org/resources. ^2^ Antibody activity fades after several months at −80 °C.

**Table 3 cells-10-00072-t003:** Staining combinations.

Double and Triple Stainings			Species
*Astrocyte selectivity*			
GFAP	Rab6A		mouse
SOX9			
GFAP	Rab6A		human
*Astrocyte specificity*			
Iba1	Rab6A		mouse
CNPase			
NG2			
*Pan-astrocyte selectivity*			
GFAP	Rab6A	GS	mouse
		Aldh1L1	

**Table 4 cells-10-00072-t004:** Classification of astrocyte types I–IV—subcellular distribution, morphology, number, and size of Rab6A^+^ grains within the individual cell.

Type	Morphology, Number, and Size of Rab6A^+^ Grains	Subcellular Distribution
I	- evenly sized - mean area: 2.5 µm^2^ - mean number *: 123	- evenly distributed all over the cell - present in perinuclear cytoplasm, stem processes, PAPs - at the cell boundary, throughout the glial territory
II	- heterogeneous size - mean area: 3.0 µm^2^ - mean number *: 104	- partly appearing in clusters - present in perinuclear cytoplasm, stem processes, PAPs - at the cell boundary, throughout the glial territory
III	- size of grains in perinuclear cytoplasm or at the cell boundary may be double or multiple of those nearby - evenly sized in stem processes, PAPs - mean area: 2.6 µm^2^ - mean number *: 70	- uneven distribution in perinuclear cytoplasm and at cell boundary - evenly distributed in stem processes and PAPs - may be only sparsely present in processes
IV	- large size, lumps - mean area: 4.3 µm^2^ - mean number *: 9	- organized in cluster(s) on one side of nucleus - sparsely present if at all, in stem processes and PAPs, thus hardly present throughout the glial territory

* number of Rab6A^+^ puncta/ROI (see Appendix A).

## Data Availability

No new data outside those presented in this study were created or analyzed. Data sharing is not applicable to this article.

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
