# Peer review of "Rab6A as a Pan-Astrocytic Marker in Mouse and Human Brain, and Comparison with Other Glial Markers (GFAP, GS, Aldh1L1, SOX9)"

_cells, 2021, doi:10.3390/cells10010072_

Round 1

Reviewer 1 Report

The manuscript “Rab6A as a pan-astrocytic marker in mouse and  human brain, and comparison with other glial markers (GFAP, GS, Aldh1L1, SOX9)” by Melzer and colleagues proposes Rab6A as a novel pan-marker for astrocytes both in mouse and human brains.

Findings are interesting, especially considering the lack of markers to specifically label astrocytes in different species. The manuscript reports clear and comprehensive experimental evidences to support the conclusions drawn by the authors. I have only one minor comment to the authors:

  • Line 215: the authors mention “data not shown” about some aspecific staining for Rab6A observed in microglia cells. Because the major point of the paper is to offer a broad overrview about Rab6A stainings, I would show ALL data, which are relevant for readers to interpret their experiments.

Reviewer 2 Report

In this manuscript Melzer and colleagues examine the expression pattern of the small GTPase RAB6A in mouse and human brain samples by immunofluorescence. They report a diffuse granular staining and, by combined co-staining, conclude that RAB6A is exclusively expressed in astrocytes, which display distinct staining patterns. Therefore, the authors propose Rab6A as a ‘global determinant of astrocyte identity’.

This study has a circumscribed focus and adopts exclusively a descriptive approach limited in mouse to part of the encephalon in naïve conditions. Although the authors present extensive and blinded quantification, relevant methodological issues limit the strength of the provided evidence in support to the authors’ conclusions.

Major concerns

  • Methodological concerns and strength of results

This reviewer appreciates the efforts to be unbiased and accurate. However, cell classification as positive/negative appears subjective ie based on a qualitative judgement of the observer and not on quantitative and reproducible measures. This is linked to the limitations of the imaging approach adopted that does not rely on what is nowadays considered as a standard methodology, ie confocal microscopy. Confocal analysis would allow discriminating the actual spatial relationships among differently labelled pixels, 3D reconstructions to show the precise localization of relevant structures within cells, and, of relevance, quantifications of pixel co-labelling in 3D across individual optical sections. To this reviewer’s opinion, a confocal analytic level as declined above is essential to support the authors’ conclusion on a topic that would benefit of superresolution microscopy or EM to be fully resolved.

Moreover, it remains unclear what was/were the criterion/a adopted to classify cells as positive or negative, for instance in conjunction with anti-GFP staining: staining contiguity? Costaining of pixels? The definition of clear objective criteria is essential to grant data reproducibility and therefore required.

Furthermore, and most relevant, the demonstration that RAB6A is expressed or not within the cells requires a very good cytoplasmic staining to highlight that the RAB6+ structures are contained or not within relevant cells and their thick or thin processes. Cytoplasmic staining is particularly unconvincing for astrocytes- which are the focus of this study. The authors should consider other strategies to make their point (eg they could obtain and stain sections of reporter mice where scattered astrocytes are fully labelled by fluorescent molecules) or obtain and examine higher quality labelling.

Along this line, positive evidence that diffuse RAB6A staining does not belong to neuronal somata/processes. The authors allude at the negativity of neurons, refer to Sox9 negative nuclei (cells) devoid of RAB6A, and report conflicting or non-conclusive literature.

Co-staining with neuronal markers is required to conclude that RAB6A is absent from neurons.

  • Impact of the study

The authors report the interesting finding that astrocytes reacting to specific pathological conditions can be RAB6A negative. Provided that the analytical strength of the study is improved, analysis of RAB6A in different pathological models may help increasing the impact of the study.

Further remarks

  • Confirmation of main interpretation of RAB6A as localized at the TGN is required so to suggest or not TGN-independent functions

  • RAB6A is presented as a ‘global determinant of astrocyte identity’ or as a ‘pan astrocytic marker’. These are overstatements based on the reported finding and the tested territories, which do not extend to all CNS areas. Further, the term ‘determinant’ implicates some form of causality with respect to the nature/outcome of something. However, in the absence of functional studies, the role of RAB6A in astrocyte identity and functioning remains to be determined. The authors should revise these statements.

Specific remarks

Figure 1/S1, commented as ‘Rab6A+ structures also fill the numerous GFAP- peripheral astrocyte processes (PAPs) that emerge from the main processes and make up the diffuse label at low power (Fig. 1 B, D; S1A, C). Rab6A+ puncta are also localized in the perivascular glial endfeet (Fig. S4)’… Figure2: Rab6A+ chains of puncta fill the GFAP-  Fine processes that are connected to the main processes (arrows in c). See also reference to ‘complete cells’ for astrocytes based on anti-GFAP staining. No direct evidence is provided to show Rab6A+ in PAPs/astrocyte fine processes. Moreover, it is established that GFAP highlights only part of astrocyte arborization, besides the fact that is only expressed in a fraction of astrocytes in naïve conditions. Thus, the interpretation of the current findings needs to be revised.

Figure 3 (anti-GS, anti- ALDH1L1 immunolabelling) appears inconclusive to show that RAB6A+ structures are contained within GS+ or ALDH1L1astrocyte (cytoplasmic) compartments, which is very relevant in this study.

Figure 4 is only partly convincing: are there Sox9+ nuclei inside vessels? Overall lower staining levels compared to other low power?

Figure 5A Almost no RAB6A is detectable in the picture illustrating anti-Iba1 staining.

The quality of anti-RAB6A labelling should be all comparable in terms of intensity/%area of positive pixels to draw reliable conclusions on cell-type coexpression.

Figure 6 Astrocyte classification

Overall anti-RAB6 labelling appears rather different in fig IV and III compared to the other 2 pictures, suggesting distinct qualities of the staining in the neuropil at the basis of the classification. How can this possibility to excluded in support of the authors’ classification?

Reviewer 3 Report

The paper by Mezler et al deals with Rab6A, a new pan astrocytic marker in mouse and human brains. It is a histological study of the spatial distribution of this protein in the adult brains and its comparison to distribution of GFAP, GS, Aldh1L1 and SPX9.  Findings will be of interest to the broad neuroscience readership. The paper is well executed and well written. There are two suggestions/concerns that need to be addressed:

  1. The study uses only a single Rab6A antibody. Normally, in study like this, one would expect the use of at least two different antibodies raised in two different species. However, to all practicality, as the minimum, the authors need to discuss this issue and expand their discussion when comparing their results to those of others and include explicit comparison of antibodies used in various studies.
  2. We need the working definition of non-reactive and reactive astrocytes in this study to be clearly stated, especially that such definitions in literature are skewed.

Round 2

Reviewer 2 Report

The authors have addressed the points formerly raised satisfactorily. In particular, they have detailed staining conditions, provided analytic specifics and clarified key criteria for double positive cell identification. On the whole, this new information with further elements provided in the rebuttal has strengthen the presented evidence and resolved the main methodological concerns of this reviewer.

The authors also refer to deconvolution algorithms which are reported to be applied in Figure S9. It remains unclear whether this is the only case where deconvolution was applied. The authors are also kindly requested to add a few details/references on this methodology in the Methods section.
